# Mutualism in disguise: A mosquito parasite with mixed transmission mode displays mutualistic traits promoting oogenesis

Maxime Girard[1], Mathieu Laÿs[1], Edwige Martin[1], Laurent Vallon[1], An-nah Chanfi[1], Mélanie Bretton[1], Aurélien Vigneron[1], Séverine Balmand[2], Patricia Luis[1], Anne-Emmanuelle Hay[1], Claire Valiente Moro[1], Guillaume Minard[1]*

1 Universite Claude Bernard Lyon 1, Laboratoire d'Ecologie Microbienne, UMR CNRS 5557, UMR INRAE 1418, VetAgro Sup, Villeurbanne, France, 2 INSA Lyon, INRAE, BF2I, UMR 203, Villeurbanne, France

* guillaume.minard@univ-lyon1.fr

## Abstract

Mutualistic traits are frequently associated with vertically transmitted symbionts, in part because repeated interactions can align host and symbiont fitness. However, how such traits emerge in symbionts combining vertical and horizontal transmission remains unclear. Here we show that *Ascogregarina taiwanensis*, previously described as a weak horizontally transmitted parasite of the Asian tiger mosquito (*Aedes albopictus*), also displays mutualistic traits that enhance mosquito reproduction. Infected females show improved embryogenesis and an extended egg-laying period, while most pseudo-vertically transmit oocysts to their progeny at oviposition sites. This interaction ultimately produces larger larvae that are more frequently infected by *As. taiwanensis*. Dual transcriptomic analyses further reveal that early oogenesis in infected females involves increased nitrogen metabolism in both partners, enhanced detoxification of blood waste, and activation of egg development pathways. These changes improve assimilation of blood proteins essential for egg production. Together, our results illustrate how physiological coupling during reproduction, combined with mother-biased transmission, can generate mutualistic traits within an interaction that also retains parasitic features, blurring the boundary between parasitism and mutualism.

## Author summary

Mosquitoes host a wide diversity of microorganisms that can profoundly influence their biology, and their effects on mosquito physiology and reproduction are often more complex than expected. In this study, we examined the interaction between the Asian tiger mosquito *Aedes albopictus*, a major invasive disease vector, and its common gregarine parasite *Ascogregarina taiwanensis*. We show that this parasite spreads in two ways: unrelated interindividual transmission

---

which permits unrestricted use, distribution, and reproduction in any medium, provided the original author and source are credited.

**Data availability statement:** Data are available at https://zenodo.org/records/14899302. Raw transcriptomic data have been deposited at the European Nucleotide Archive under the accession number ERS23447863.

**Funding:** This work was supported by the Centre National de la Recherche Scientifique, website: programmes.insu.cnrs.fr/ec2co, year: 2020, Grant number: CNRS EC2CO Interasco, holder: G.M and Agence Nationale de la Recherche, website: anr.fr, year: 2024, Grant number: ANR-23-CE35-0014, holder: G.M. The Ph.D. of M.G. was supported by a French ministerial fellowship from the Ecology, Evolution, Microbiology, and Modeling doctoral school of the University Claude Bernard Lyon 1. We thank the Master of Microbiology programs of the University Claude Bernard Lyon 1 and the University Clermont Auvergne, in which M.L., M.B., and A-n.C. carried out their traineeships. The funders had no role in study design, data collection and analysis, decision to publish, or preparation of the manuscript.

**Competing interests:** The authors have declared that no competing interests exist.

can occur through breeding sites, but its transmission is also biased toward the offspring of infected females because parasites are released into the water at egg-laying sites. Rather than acting solely as a cost to the host, infection was associated with changes in reproductive physiology: parasitized females produced larger eggs, laid them over a longer period, and gave rise to larger larvae, despite ingesting similar amounts of blood as uninfected females. Transcriptomic and physiological analyses indicate that these effects are linked to enhanced processing of blood-derived nutrients, particularly nitrogen, during oogenesis. These findings suggest that physiological responses to infection during reproduction can generate host benefits that also favor parasite transmission, illustrating how mutualistic traits may emerge as by-products of adaptation within predominantly parasitic interactions.

## Introduction

Host-microbe interactions are commonly classified along a continuum ranging from parasitism to mutualism, depending on their negative, neutral, or positive effects on each partner [1]. While environmental conditions and genetic backgrounds modulate these interactions [2], the degree of dependency between partners also plays a crucial role in shaping the trajectory of the relationship [3]. Interdependency between partners is influenced by the mode of transmission, but also by the degree of reciprocal adaptation and the genetic context in which interactions occur; vertical transmission often reinforces such dependency by aligning host and symbiont fitness [4–6].

Many mutualists and parasites are thought to have evolved from free-living ancestors, some of which have transitioned into obligate symbionts over generations [7]. Empirical evidence suggests that parasitism is often the initial step following a free-living lifestyle [7–9]. For example, the free-living, photosynthetic ancestors of Apicomplexa lost chloroplast genes over the course of evolution and shifted from autotrophy to heterotrophy [10]. To compensate for the loss of essential metabolic capabilities, Apicomplexa became highly dependent on the animal hosts they parasitize for nutrient acquisition. In some cases, interactions between hosts and environmental microbes became so critical that mutualistic relationships evolved. For instance, certain *Pantoea* species transitioned from free-living soil bacteria to obligate mutualists inhabiting specialized midgut crypts in the stinkbug *Plautia stali* [11]. This association is maintained overtime through vertical transmission, *i.e.,* females coat the egg masses with secretions containing symbionts, which are then ingested by the newly hatched nymphs [12].

Inter-individual transmission of microorganisms is a major force shaping the nature of host–microbe interactions [13]. Vertical transmission, in particular, promotes tighter host–microbe associations and can lead to reduced virulence and the evolution of mutualistic traits that enhance the reproduction of both partners [7]. In contrast, horizontal transmission can decouple symbiont fitness from that of the host, potentially favoring parasitic strategies, although such outcomes depend strongly on

host–parasite adaptation and genetic homogeneity within interacting populations [1,14]. For example, substantial evidence suggests that the aphid symbiont *Serratia symbiotica* was originally a horizontally transmitted parasite that evolved mutualistic characteristics under the selective pressure of vertical transmission [15,16]. Similarly, the virulence of the free-living alga *Symbiodinium microadriaticum* has been shown to decrease under experimental vertical transmission in jellyfish hosts [17]. Together, empirical studies and controlled experiments demonstrate that vertical transmission often reduces microbial virulence and may promote the emergence of mutualistic traits, particularly when host and symbiont experience repeated interactions and opportunities for reciprocal adaptation [18,19]. However, direct observations of an ongoing transition from parasitism to mutualism driven by vertical transmission remain scarce in the literature.

The Asian tiger mosquito, *Aedes albopictus*, is the most invasive vector of vertebrate pathogens, and its global spread is a major concern for public health policies [20]. Field populations of *Ae. albopictus* are commonly and heavily parasitized by *Ascogregarina taiwanensis*, a low-virulence Apicomplexan entomoparasite [21]. This parasite negatively affects host survival and reproduction under nutrient-deficient conditions [22] or when infection loads are high [23]. The outcome of this interaction is further shaped by genotype-by-genotype effects between host and parasite [24,25], and *As. taiwanensis* has been suggested to contribute to the regulation of mosquito populations [26]. However, it has also been suggested that *As. taiwanensis* may have contributed to the competitive success of *Ae. albopictus* over other mosquito species during the invasion of new habitats—such as breeding sites also colonized by *Ae. aegypti* [27]. This may be due to the higher virulence of *As. taiwanensis* toward *Ae. aegypti* larvae, while exerting limited impact on the fitness of its natural host [28]. Consequently, *As. taiwanensis* may have either negative or indirectly beneficial effects on *Ae. albopictus*.

Infection by *As. taiwanensis* occurs when larvae ingest oocysts in breeding water, after which the parasite develops in the midgut and Malpighian tubules of pupae and adults [29]. It is then disseminated by adult mosquitoes via a supposedly mixed transmission mode [30]. Briefly, horizontal transmission occurs when emerging adults release meconium in aquatic habitats, or when infected individuals defecate or die in water. Several studies have also suggested a potential pseudo-vertical transmission (*i.e.,* maternal transmission bias toward post hatch symbiont acquisition) via egg smearing from females to their offspring [23,30], although this has yet to be clearly demonstrated.

In this study, we investigated whether *As. taiwanensis* transmission is biased toward the offspring of infected females through pseudo-vertical mechanisms, while co-occurring with horizontal transmission. We found that this transmission route is indeed possible, even though it co-occurs with horizontal transmission in breeding sites. Given the close link between female reproduction and parasite transmission, we then examined the impact of infection on mosquito oogenesis, fertility, fecundity, and offspring quality. We showed that, although parasitized and non-parasitized mosquitoes ingest similar amounts of blood, infection promotes oogenesis in females, resulting in larger eggs laid over an extended oviposition period and ultimately yielding larger larvae. Dual RNA-seq conducted across reproductive stages revealed that pathways involved in the assimilation of blood-derived nutrients—particularly nitrogen—were upregulated in both the parasite and the mosquito. In addition, parasitized females exhibited a more efficient turnover of protein digestion by-products during early oogenesis and greater protein assimilation toward the end of oogenesis.

## Results

### *As. taiwanensis* is pseudo-vertically transmitted through the aquatic habitat

To understand how *As. taiwanensis* persists across generations, we first examined its prevalence in females at different stages of the reproductive cycle. The proportion of females harboring oocysts remained high and stable in unmated (UNMAT; 81.2%), mated (MATED; 87.5%), and recently blood-fed females (86.7%; 1 day after blood meal, 1DABM) (Fig 1a and Table 1). However, oocyst prevalence sharply dropped during late oogenesis (3DABM) to an average of 40%, suggesting parasite release during oviposition (Fig 1b).

To test whether females transmit the parasite to their offspring via the aquatic habitat, we measured the prevalence of *As. taiwanensis* trophozoites in larvae. When larvae developed in the same water in which their parasitized mothers laid

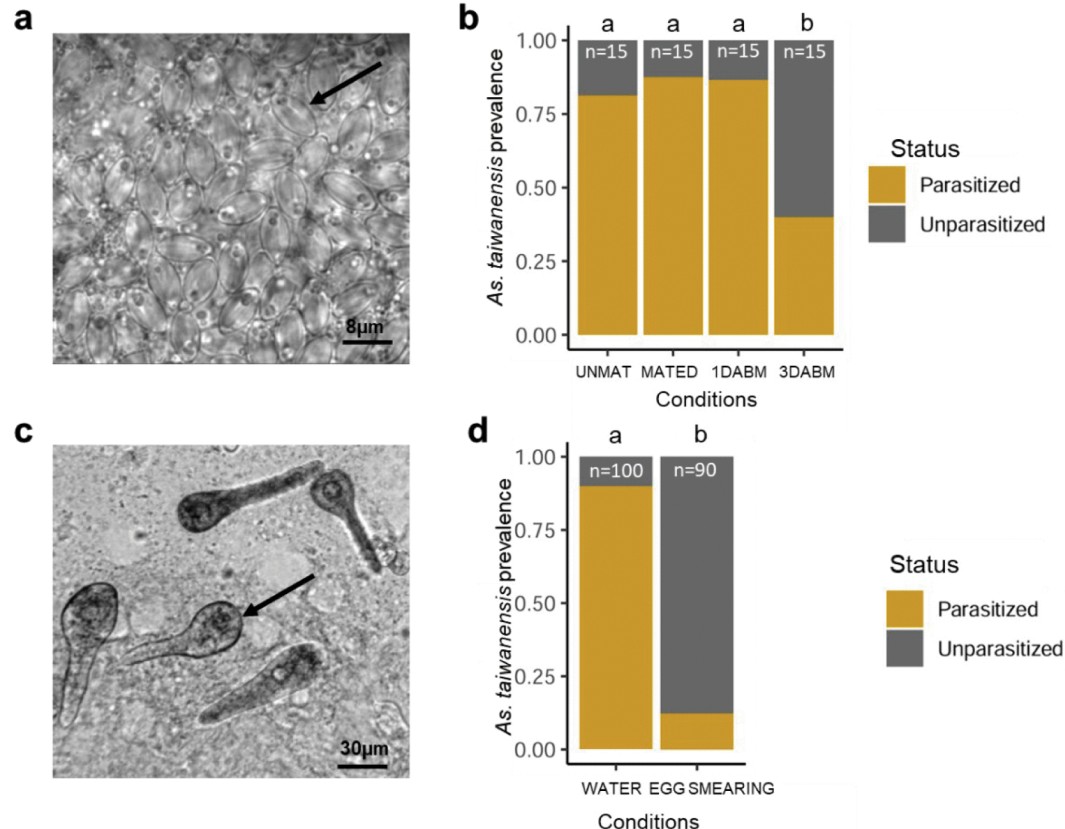

**Fig 1. Prevalence of *As. taiwanensis* in females over life stages and vertical transmission. (a)** The prevalence of *As. taiwanensis* oocysts was measured in females. The image was acquired using light microscopy at ×1000 magnification; the black arrow indicates a single oocyst. **(b)** Prevalence of oocysts in female Malpighian tubules at different physiological stages: unmated (UNMAT), mated (MATED), one day after blood meal (1DABM), and three days after blood meal (3DABM). **(c)** The prevalence of *As. taiwanensis* was estimated in larvae in its trophozoïte. The image was acquired using light microscopy at ×400 magnification; the black arrow indicates a single trophozoite. **(d)** Vertical transmission assessed in newly hatched larvae exposed either to water from oviposition cups (WATER) or to sterile water after egg surface sterilization (EGG SMEARING). Compact letter displays indicate statistically significant differences based on post hoc Tukey HSD tests.

eggs, infection prevalence reached 89%. In contrast, only 12.2% of larvae were infected when eggs were transferred to sterile water before hatching (Fig 1c and 1d and Table 1). These results indicate that pseudo-vertical transmission occurs predominantly via oocysts released in water, rather than through egg smearing. Because egg smearing appeared weak, a second experiment was conducted to determine whether oocysts were firmly attached to the eggs. While larval infection prevalence reached 14.3% in larvae emerging from directly flooded eggs, no infected larvae were observed from rinsed eggs, suggesting that the parasite can be easily detached when eggs are immersed in water (S1 Fig).

### Parasitized females produce larger eggs over an extended laying period and give rise to larger larvae

*As. taiwanensis* colonizes the Malpighian tubules—organs involved in excretion and osmoregulation, particularly active during blood meal digestion. Parasitized females harbor a slightly larger abdomen surface than their unparasitized con-specific ($1.52 \pm 0.220 mm^2$ vs. $1.46 \pm 0.367 mm^2$, t = 2.23, p = 0.0253; S2 Fig). Following blood feeding, both parasitized and unparasitized females ingested similar blood volumes, as inferred from comparable abdomen width (Fig 2a and Table 1).

**Table 1. Statistical analysis summary.**

| Experiment | Model | Response variable | Fixed effect | X²/ F | p-Value |
|---|---|---|---|---|---|
| Adult Prevalence | GLM | Prevalence | Female conditions | 10.46 | **0.015*** |
| Transmission | GLMM | Prevalence | Transmission conditions | 40.96 | **<0.0001*** |
| Blood intake | LMM | | Blood meal | 138.44 | **<0.0001*** |
| | | | Parasitism status | 0.01 | 0.91 |
| | | | Blood meal x Parasitism status | 0.04 | 0.85 |
| Oogenesis | LMM | Primary chamber area | DABM | 4154.67 | **<0.0001*** |
| | | | Parasitism status | 18.17 | **<0.0001*** |
| | | | DABM x Parasitism status | 4.90 | 0.18 |
| | | Yolk area | DABM | 3987.87 | **<0.0001*** |
| | | | Parasitism status | 17.87 | **<0.0001*** |
| | | | DABM x Parasitism status | 2.19 | 0.53 |
| Egg laying | GLMM | Egg laying dynamic | DABM | 590.41 | **<0.0001*** |
| | | | Parasitism status | 4.09 | **0.043*** |
| | | | DABM x Parasitism status | 22.26 | **<0.0001*** |
| Egg hatching | GLMM | Hatching rate | Desiccation | 47.77 | **<0.0001*** |
| | | | Parasitism status | 4.78 | **0.029*** |
| | | | Desiccation x Parasitism status | 0.01 | 0.93 |
| Larval size | LMM | Larval size | Parasitism status | 13.63 | **0.0002*** |
| Protein, Uric acid and Uricase | LMM | Protein content | Stages | 6.74 | **<0.0001*** |
| | | | Parasitism status | 22.79 | **<0.0001*** |
| | | | Stages x Parasitism status | 1.61 | 0.19 |
| | | Uric acid content | Stages | 27.52 | **<0.0001*** |
| | | | Parasitism status | 36.51 | **<0.0001*** |
| | | | Stages x Parasitism status | 0.98 | 0.48 |
| | | Uricase activity | Stages | 21.13 | **<0.0001*** |
| | | | Parasitism status | 1.13 | 0.29 |
| | | | Stages x Parasitism status | 4.79 | **0.004*** |

*p ≤ 0.05, ** p ≤ 0.01, ***p ≤ 0.001, Bold: statistically significant *p*-values.

Despite similar blood intake, parasitized females showed enhanced oogenesis. Primary egg chamber area increased by 7.8% at 2DABM (t = 3.31, p = 0.001**) and 6.5% at 3DABM (t = 2.94, p = 0.004**), while yolk (vitellus) area increased by 19.8% at 1DABM (t = 2.66, p = 0.001**), 8.1% at 2DABM (t = 2.03, p = 0.02*), and 8.6% at 3DABM (t = 2.64, p = 0.01*) (Fig 2b–2d).

Egg laying was delayed in parasitized females. In both groups, most eggs were laid at 4 days after blood meal (DABM), accounting for 72.6% and 93.0% of eggs in parasitized and unparasitized females, respectively (Figs 2e and S3). However, a temporal delay was evident, as parasitized females laid significantly fewer eggs at both 3 and 4 DABM (3 DABM: z = −2.23, p = 0.026*; 4 DABM: z = −2.14, p = 0.032*). The total number of eggs laid did not differ significantly between groups (z = −1.46, p = 0.144), although it was slightly lower in parasitized females. Hatching success was comparable between groups, regardless of whether eggs were fresh or stored (Fig 2f). Importantly, larvae hatching from eggs laid by parasitized females were significantly larger (+8.4%) than those from unparasitized females (Fig 2g and Table 1), suggesting a developmental advantage potentially linked to improved maternal resource allocation.

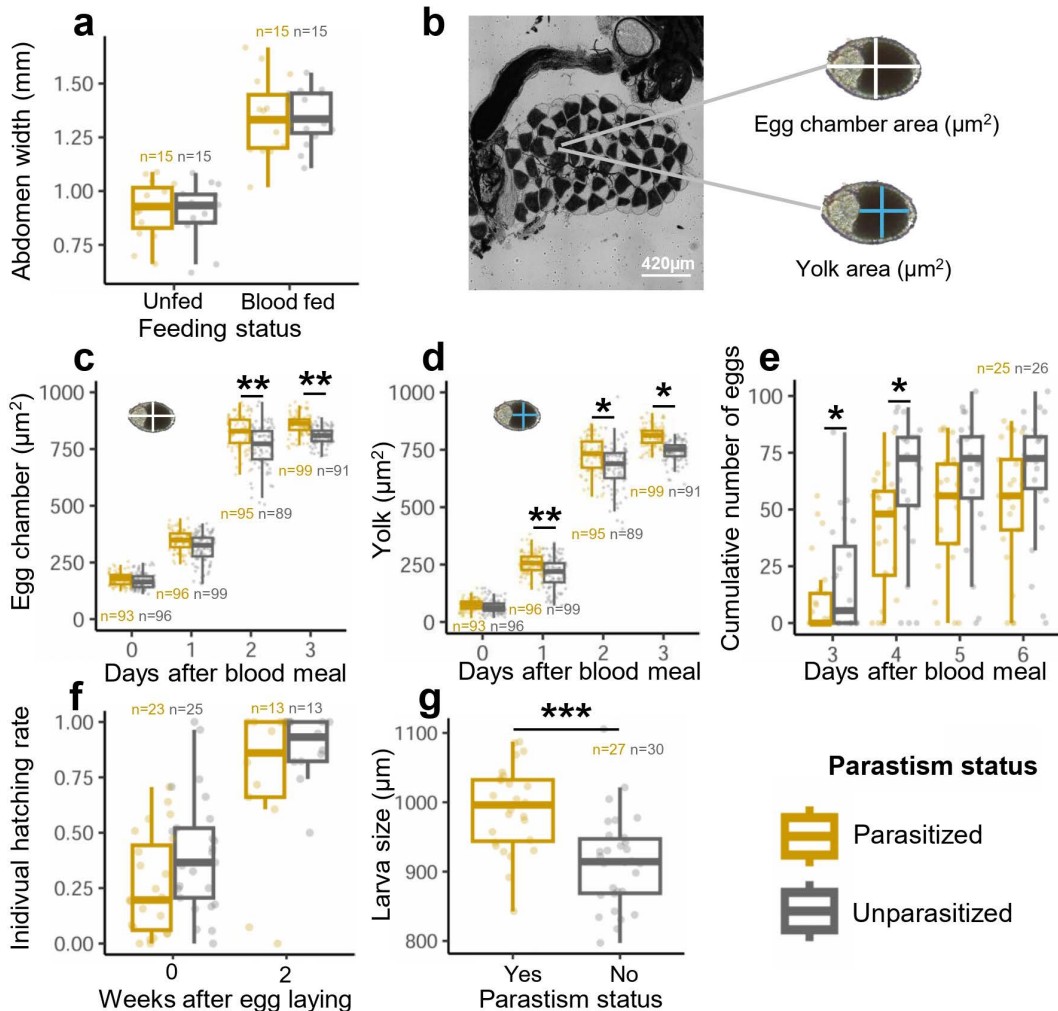

**Fig 2. Impact of *As. taiwanensis* on female blood feeding, oogenesis, egg laying, egg hatching and larval size. (a)** Abdomen width was measured in parasitized and unparasitized females, before (Unfed) and after a blood meal (Blood fed). **(b)** Representative image of primary follicle chamber and yolk area in a parasitized female one day after blood feeding (1 DABM), visualized using light microscopy (×100 magnification). **(c)** Area of primary follicle chambers and **(d)** yolks was quantified daily from 0 to 3 DABM in parasitized and unparasitized females. **(e)** Egg-laying dynamics were monitored daily from 3 to 6 DABM in parasitized and unparasitized females and represented as cumulative egg numbers. **(f)** Hatching success was measured with (2 weeks) and without (0 weeks) a desiccation period in both parasitized and unparasitized groups. **(g)** Larval size was assessed after hatching for parasitized and unparasitized individuals. Asterisks indicate significant differences (*p ≤ 0.05, **p ≤ 0.01, ***p ≤ 0.001; Tukey HSD post hoc test). Biorender was used to generate the figure (Licence number TW29FC9MJM). Created in BioRender. Girard, M. (2026) https://BioRender.com/lyuz2tc.

## Parasitism and reproductive stage shape the mosquito transcriptome

We next examined transcriptomic changes in female mosquitoes across reproductive stages and infection status (UNMAT, MATED, 1DABM, and 3DABM). Of 36,525 transcripts, 30,827 were retained for downstream analysis. PCA revealed that both female life stage and parasitism status significantly structured gene expression profiles (Fig 3a), accounting for 49.6% of total variance: 34% by life stage (F = 7.20, p = 0.001***), 3.8% by infection status (F = 2.39, p = 0.067), and 11.8% by their interaction (F = 2.51, p = 0.012*).

Differentially expressed genes (DEGs) between parasitized and unparasitized mosquitoes were analyzed (Fig 3b). Each DEG represents a transcript for which the expression abundance is significantly higher either in parasitized or

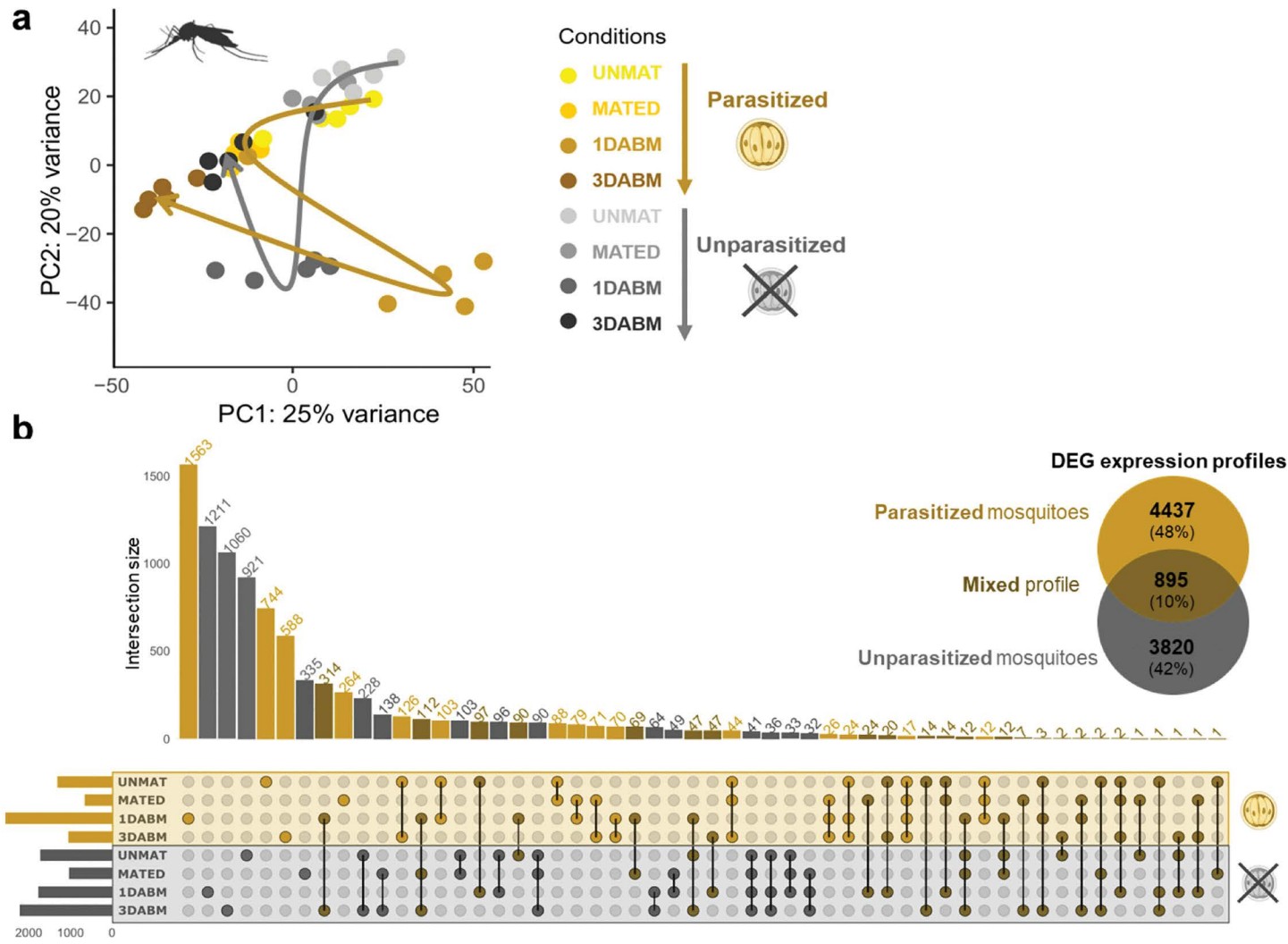

**Fig 3. Transcriptomic impact of *Ascogregarina taiwanensis* on *Ae. albopictus* females: Principal Component Analysis and differentially expressed genes (DEGs). (a)** Principal Component Analysis (PCA) of transcriptome profiles from 5 parasitized and 5 unparasitized females at four physiological stages: unmated (UNMAT), mated (MATED), one day after blood meal (1DABM), and three days after blood meal (3DABM). Each dot represents the transcriptome of an individual female. **(b)** A Venn diagram and an UpSet plot summarize the number and overlap of DEGs between parasitized and unparasitized mosquitoes across female conditions. A total of 9,152 DEGs were identified using pairwise comparisons and Wald tests with Benjamini-Hochberg correction. Intersections are shown as columns. Colors indicate genes specific to parasitized mosquitoes (yellow), unparasitized mosquitoes (grey), or shared between both conditions (dark yellow). Female physiological stages are color-coded and labeled as UNMAT, MATED, 1DABM, and 3DABM. Biorender was used to generate the figure (Licence number TW29FC9MJM). Created in BioRender. Girard, M. (2026) https://BioRender.com/lyuz2tc.

unparasitized mosquitoes. In total, 9,152 differentially expressed genes (DEGs) were identified (Fig 3b), with 4,437 and 3,820 genes specific to parasitized and unparasitized females, respectively, and 895 shared. The greatest divergence was observed at 1DABM, with 2,626 DEGs, including 1,563 specific to parasitized females.

## Parasitized mosquitoes upregulate genes related to development and protein metabolism after blood feeding

GO enrichment analysis showed that parasitism drives distinct gene expression programs depending on reproductive stage (Fig 4). In UNMAT and MATED females, parasitized individuals overexpressed genes involved in macromolecule,

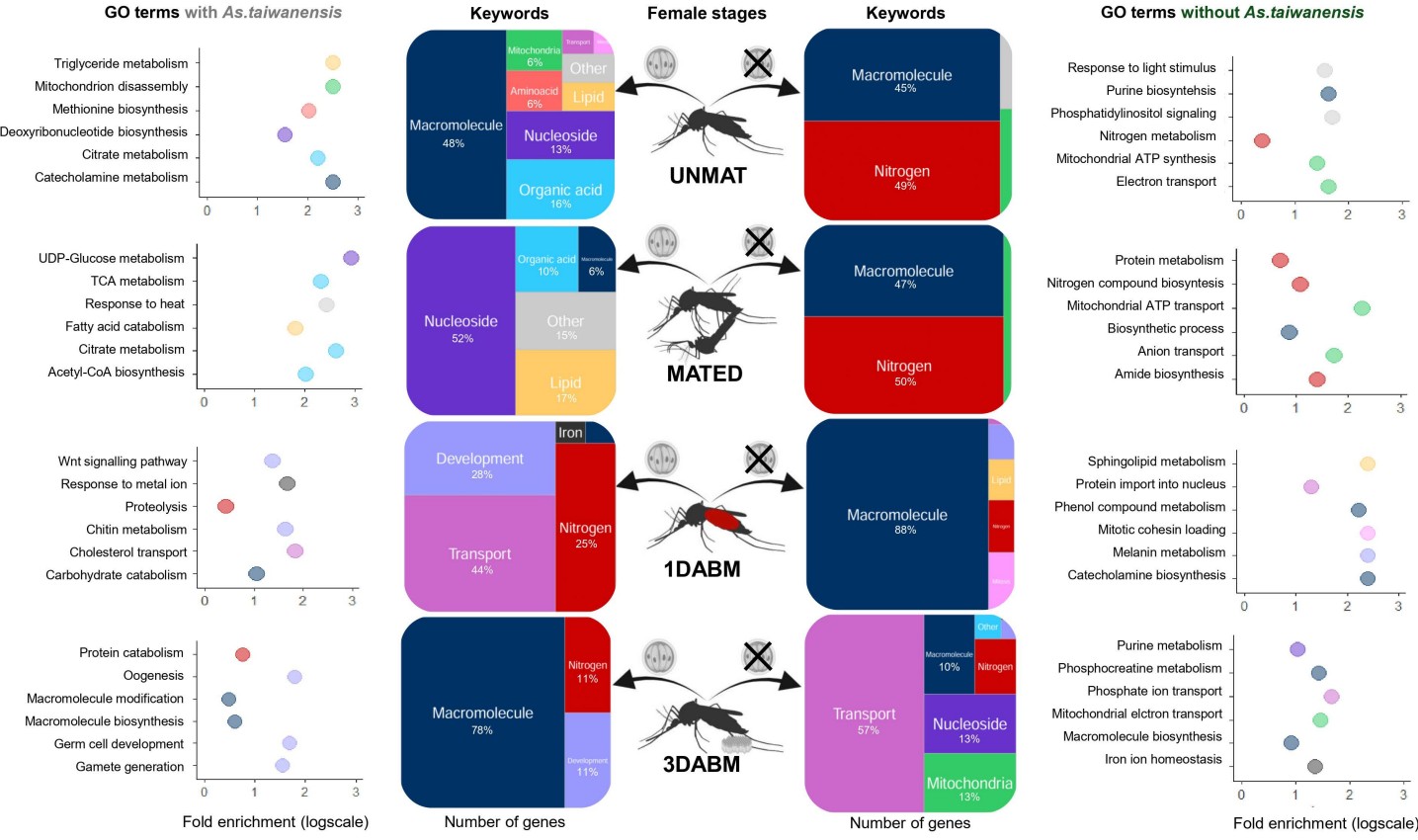

**Fig 4. Functional annotation of differentially expressed genes (DEGs) between parasitized and unparasitized *Ae. albopictus* females across life stages.** Treemaps display enriched functional categories summarizing Gene Ontology (GO) terms associated with DEGs at four female physiological stages: unmated (UNMAT), mated (MATED), one day after blood meal (1DABM), and three days after blood meal (3DABM). The surface area of each keyword reflects the proportion of associated DEGs. For each stage and infection status, the fold enrichment scores of the top six GO terms are shown (log scale). Percentages of genes involved in each functional category are displayed when exceeding 1%. Biorender was used to generate the figure (Licence number TW29FC9MJM). Created in BioRender. Girard, M. (2026) https://BioRender.com/lyuz2tc.

lipid, nucleoside, and organic acid metabolism. Enrichment of GO terms associated with lipid (*e.g.,* fatty acids) and nucleoside (*e.g.,* UDP-glucose metabolism) processing was more pronounced in MATED parasitized individuals (52% and 17% of the total number of genes respectively). Conversely, transcriptome profiles of unparasitized mosquitoes were similar during the UNMAT and MATED life stages. They upregulated functions related to nitrogen (*e.g.,* nitrogen/protein metabolism, amide and nitrogen compound biosynthesis), macromolecule (*e.g.,* biosynthesis of compounds such as purine) metabolism as well as mitochondrial (*e.g.,* ATP synthesis, anion, electron and ATP transport) activity.

After blood feeding, transcriptomic differences intensified. At 1DABM and 3DABM, parasitized females overexpressed genes involved in development (e.g., wnt signaling pathway, chitin metabolism, oogenesis germ cell development and gamete generation) and nitrogen metabolism (e.g., proteolysis/protein catabolism; Fig 4). In contrast, DEGs in 1DABM unparasitized females were dominated by macromolecule metabolism (88%).

Iron metabolism was also more active in parasitized 1DABM females (2% of DEGs) than in 3DABM unparasitized females (<1%). A focused analysis of enriched metabolic pathways in 1DABM parasitized females (S1 Table) revealed upregulation of amino acid metabolism, detoxification (glutathione, cytochrome P450), lipid catabolism, hormonal regulation, vitamin metabolism (ascorbate) and energy (pyruvate) pathways.

Notably, several serine protease-like genes involved in protein catabolism showed very high fold changes (129.7, 55.4, 38.6). Genes for glutamine synthetase and glutamate dehydrogenase-like enzymes (GS/GOGAT cycle) were also highly upregulated (fold changes: 6.9, 6.1), as were detoxification genes like glutathione S-transferase-like (fold change: 116.5) and peroxiredoxin (8.1). Hydroxysteroid 17-beta dehydrogenase 11 (HSD17B11), involved in lipid metabolism and hormonal regulation, was upregulated by 20.8-fold.

## The transcriptome of *As. taiwanensis* remains mostly stable across mosquito life stages

Given the strong effect of parasitism on the mosquito transcriptome, we also examined the transcriptional dynamics of the symbiont *As. taiwanensis* across female life stages. PCA conducted on the parasite gene expression profiles (based on 1,679 transcripts retained after abundance estimation and normalization from 4,676 identified) revealed no significant global variation among life stages (F = 1.10, p = 0.149; Fig 5a). This suggests that the overall *As. taiwanensis* transcriptome remains relatively stable across conditions.

Despite the absence of a major global pattern, differential gene expression (DGE) analysis identified 438 DEGs across life stages. Among them, 60 genes were shared between conditions, while stage-specific DEGs ranged from 68 to 130 (Fig 5b).

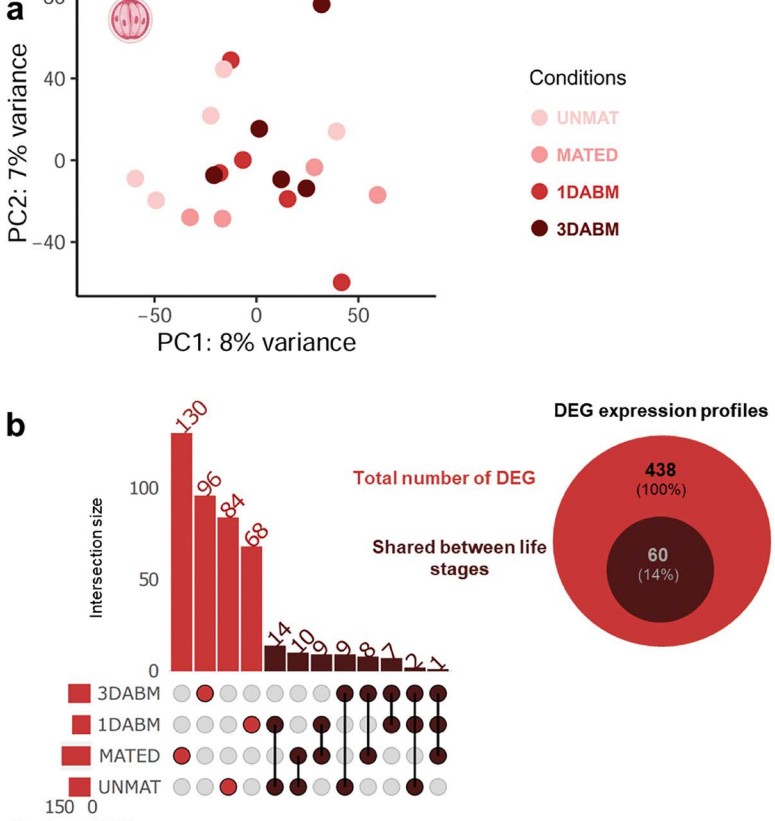

**Fig 5. Principal Component Analysis and differentially expressed genes (DEGs) in the *Ascogregarina taiwanensis* transcriptome. (a)** Principal Component Analysis (PCA) of de novo transcriptome profiles from *As. taiwanensis* within 5 parasitized *Ae. albopictus* females across four physiological stages: unmated (UNMAT), mated (MATED), one day after blood meal (1DABM), and three days after blood meal (3DABM). Each dot represents the parasite transcriptome associated with an individual host. **(b)** A Venn diagram and UpSet plot summarize the number and overlap of DEGs identified across stages. A total of 438 DEGs were detected using pairwise comparisons and Wald tests with Benjamini-Hochberg correction. Among them, 378 DEGs were specific to a single life stage (red), while 60 were shared across multiple stages (dark red).

## Transcriptomic variation in *As. taiwanensis* during mosquito reproduction

To provide a better overview of the mechanisms involved in host-parasite interactions, functional annotations of the *As. taiwanensis* DEG were also performed (Fig 6). The main GO term keywords associated with genes expressed in *As. taiwanensis* were related to macromolecule, nitrogen and nucleoside metabolism. Nitrogen metabolism, which was particularly abundant during the early phase of mosquito blood digestion and oogenesis (*i.e.,* 1DABM with 28% of the genes), was consistently the second most important keyword across female life stages. This includes genes involved in protein

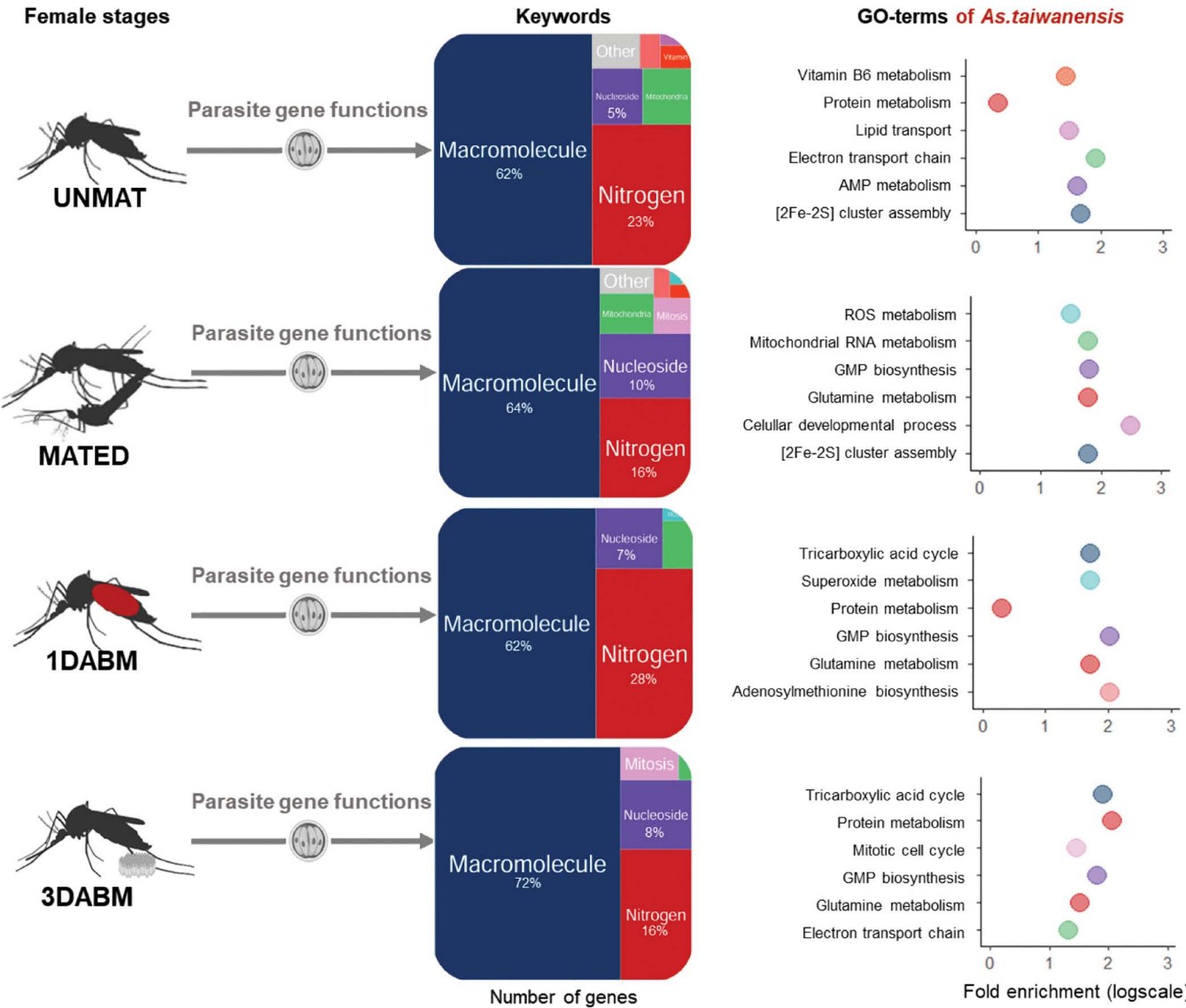

**Fig 6. Functional annotation of *Ascogregarina taiwanensis* transcriptome across mosquito female life stages.** The *As. taiwanensis* transcriptome associated with *Ae. albopictus* females was functionally annotated at four physiological stages: unmated (UNMAT), mated (MATED), one day after blood meal (1DABM), and three days after blood meal (3DABM). GO terms were simplified into functional keywords and visualized as treemaps. The surface area of each keyword reflects the proportion of DEGs it aggregates (log scale). For each life stage, the top six enriched GO terms are shown with their fold enrichment scores (logarithmic scale). Percentages of genes involved in each keyword are displayed when exceeding 1%. Biorender was used to generate the figure (Licence number TW29FC9MJM). Created in BioRender. Girard, M. (2026) https://BioRender.com/lyuz2tc.

metabolism, particularly glutamine metabolism (Fig 6). For instance, the *glutamine amidotransferase* like, which facilitates the extraction of ammonia from glutamine, was identified. The arginine biosynthesis pathway was also notably enriched in both 1DABM and 3DABM conditions (S2 Table). In contrast, the C5-branched dibasic acid metabolic acid pathway, which contributes to the synthesis of precursors for branched-chain amino acids such as leucine, isoleucine and valine, was only enriched in 1DABM. Additionally, genes related to macromolecule GO-terms were more abundant in blood-fed females (1DABM and 3DABM), especially those involved in the tricarboxylic acid cycle.

To further explore potential stage-specific functional shifts in *As. taiwanensis*, we annotated the identified DEGs (Fig 6). The most enriched GO terms were related to macromolecule metabolism, nitrogen metabolism, and nucleoside metabolism. Nitrogen metabolism was particularly prominent at 1 day after blood meal (1DABM), representing 28% of DEGs at this stage. This included genes involved in protein and glutamine metabolism, such as a *glutamine amidotransferase*-like gene, which facilitates ammonia extraction from glutamine. Additionally, the arginine biosynthesis pathway was significantly enriched in both 1DABM and 3DABM (S2 Table). In contrast, the C5-branched dibasic acid metabolism pathway—implicated in the synthesis of branched-chain amino acid precursors (e.g., leucine, isoleucine, valine)—was specifically enriched at 1DABM.

Genes associated with macromolecule-related GO terms, particularly those involved in the tricarboxylic acid (TCA) cycle, were more abundant in blood-fed females (1DABM and 3DABM), suggesting a metabolic shift aligned with the host's reproductive physiology.

### A potential mutualistic role of *As. taiwanensis* in blood resource assimilation

To test whether *As. taiwanensis* contributes to enhanced resource assimilation during oogenesis, we conducted complementary experiments. Under blood dilution treatments, parasitized females were less impacted than their unparasitized counterparts. When fed blood diluted fivefold, unparasitized females failed to produce eggs, while parasitized females formed an average of 10±7 eggs (S4 Fig). These results suggest that *As. taiwanensis* may enhance nutrient assimilation, particularly of proteins, during reproduction.

To investigate this further, we measured total protein content in females across reproductive stages. In both groups, protein levels increased by ~50% at 1DABM and declined by 3DABM. However, parasitized females showed significantly higher protein levels at 3DABM, accumulating 37.8% more protein than unparasitized ones (Fig 7a).

We then quantified uric acid levels and uricase activity—two indicators of nitrogenous waste metabolism [31]—to assess whether increased protein levels in parasitized females reflected more efficient protein digestion. Both uric acid (Fig 7b) and uricase activity (Fig 7c) increased significantly between 1DABM and 3DABM, particularly in parasitized females. These findings support the hypothesis that *As. taiwanensis* enhances protein assimilation and nitrogen metabolism during blood digestion and contribute to oogenesis.

### Discussion

Empirical studies have shown that vertical transmission favors the evolution of low-virulence or even beneficial parasitic traits [1]. Mixed transmission modes—combining vertical and horizontal pathways—may also select for reduced virulence, depending on the relative importance of each route for host and parasite fitness [1,7,13]. However, increasing evidence suggests that such outcomes depend not only on transmission mode *per se*, but also on the degree of host–parasite adaptation and the genetic homogeneity of interacting populations [32–35]. The pseudo-vertical transmission of *Ascogregarina* species, however, remains poorly characterized [30]. Previous work suggested that males primarily mediate horizontal transmission by contaminating breeding water, while females disperse oocysts to new sites, mimicking vertical transmission [36]. Since *As. taiwanensis* is expelled from adult mosquitoes without replicating in the adult stage [29], our results support the hypothesis that relatively few parasites are released by females prior to oviposition, while oocyst release is enhanced during egg formation and oviposition. These oocysts efficiently infect the progeny within the

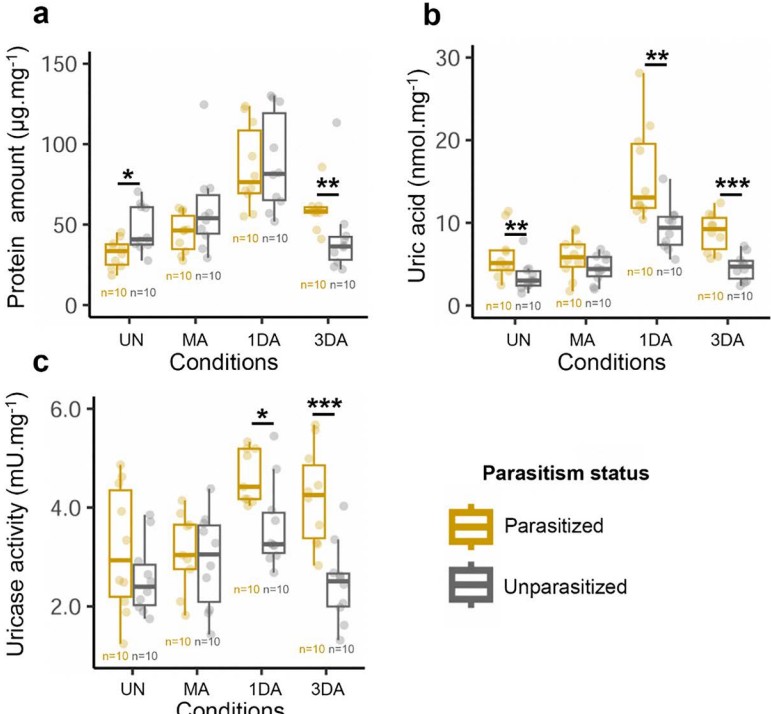

**Fig 7. Quantification of proteins, uric acid, and uricase activity in parasitized and unparasitized *Ae. albopictus* females across physiological stages. (a)** Total protein content, **(b)** uric acid concentration, and **(c)** uricase activity were measured in parasitized and unparasitized females at four life stages: unmated (UN), mated (MA), one day after blood meal (1DA), and three days after blood meal (3DA). Measurements were performed on 10 biological replicates per condition, each consisting of three pooled females and normalized to dry mass. Asterisks indicate statistically significant differences (*$p \leq 0.05$, **$p \leq 0.01$, ***$p \leq 0.001$; Tukey HSD post hoc test).

water containers. We therefore suggest that oocyst transmission by females is biased toward increasing the likelihood of transmission to their own offspring, although larvae from other females sharing the same breeding site may also become infected. The relative contributions of horizontal and vertical transmission nevertheless remain to be determined. Rather than acting as a primary driver of mutualism, this transmission ecology may reinforce stable host–parasite associations in systems where host and parasite are already well adapted to one another.

Consistent with this view, our findings indicate that *As. taiwanensis* infection is associated with improved reproductive performance in *Ae. albopictus*. Parasitized females produced larger progeny and sustained oogenesis even under limited blood resources. These benefits appear to be linked to enhanced nutrient assimilation, especially of nitrogenous compounds derived from blood proteins. Additionally, parasitized females exhibited altered oviposition behavior, distributing their eggs over a longer period than unparasitized females. This trait could increase both reproductive success and parasite transmission. *Aedes albopictus* females typically distribute their eggs among multiple containers—a bet-hedging strategy known as skip oviposition [37]. Consequently, an extended egg-laying period may enhance the dispersal of both eggs and oocysts across breeding sites.

Vertical transmission is often tightly coupled with host reproductive physiology, particularly when symbionts rely on host cellular machinery for follicle entry [38]. For instance, *Wolbachia* and *Babesia bovis* hijack vitellogenin receptors to access ovarian tissues in true bugs [39] and ticks [40], respectively. Some symbionts even enhance host resource allocation to oocytes, as seen in *Nasuia deltocephalinicola*, which increases protein transport by 20% in the rice leafhopper *Nephotettix cincticeps* [41]. While most such interactions involve direct contact with the ovary, *As. taiwanensis* remains confined to the

Malpighian tubules and does not colonize ovarian tissues. Its release alongside eggs suggests a pseudo-vertical transmission route more similar to that observed in the bacterial symbionts *Pantoea* sp. of the green stink bug *Nezara viridula* [42,43] or *Burkholderia* sp. in the oriental chinch bug *Cavelerius saccharivorus* [44]. These facultative symbionts, transmitted through egg smearing or environmental contamination, also confer fitness benefits despite lacking direct contact with reproductive organs.

Our study further supports the presence of host physiological responses to *As. taiwanensis* infection that generate outcomes consistent with mutualistic traits, particularly during blood digestion and oogenesis. In mosquitoes, blood meal digestion triggers complex endocrine responses involving the brain, fat body, midgut, and ovaries [45]. Proteins from the blood are broken down into amino acids that fuel yolk protein synthesis via the TOR pathway and promote follicle development through hormonal cascades involving insulin-like peptides and ecdysteroids [46]. Concurrently, nitrogenous waste generated from protein catabolism—uric acid, urea, or ammonia—must be either excreted or detoxified [47] via the uricotelic or trans-deamination pathways [48], with amino acids like glutamine and proline acting as key intermediates [49].

In parasitized females, genes involved in proteolysis and protein catabolism were upregulated, suggesting increased amino acid availability. Transcriptomic signatures also pointed to enhanced oogenesis hormonal signaling, including overexpression of *HSD17B11-like*, part of the steroid biosynthesis pathway. Notably, the *GS-GOGAT* enzyme complex—central to ammonia assimilation into glutamine and glutamate—was also overexpressed in parasitized females [50], supporting improved nitrogen recycling.

Although *As. taiwanensis* is not known to be metabolically active in the adult mosquito, our transcriptomic analysis revealed that it remains transcriptionally active. Genes involved in nitrogen metabolism, particularly glutamine metabolism, were highly expressed. These findings raise the possibility that the parasite utilizes excess glutamine produced by the host, possibly via its own glutamine amidotransferase. Such reciprocal use of nitrogenous compounds suggests a metabolic interplay that may benefits both partners through enhanced nitrogen turnover. However, the routes of transport at this stage remain to be investigated.

Symbiont-mediated nitrogen metabolism is well documented in phytophagous insects. Nitrogen-fixing or recycling microbes, such as diazotrophs in the long-horned beetle [51], *Buchnera aphidicola* in aphids [52], and *Ischyrobacter davidsoniae* in turtle ants [53], enable hosts to cope with nitrogen-poor diets. While mosquitoes ingest protein-rich blood (60–80 mg/mL) [54], they assimilate only a fraction of it: about 10% for oogenesis, 20% for storage, and the remainder is excreted as waste [55]. Our results suggest that *As. taiwanensis* enhances protein assimilation and nitrogen detoxification, shifting this balance toward increased reproductive output.

Beyond oogenesis, parasitized females laid eggs over a longer period and produced larger larvae—two traits known to enhance offspring survival. Extended oviposition period may enhance "skip oviposition" behavior, a bet-hedging strategy in *Ae. albopictus* where eggs are spread across multiple breeding sites [37]. Larger larvae generally outperform smaller ones in both intra- and interspecific competition, improving juvenile survival under natural conditions [56–58]. These offspring-level also benefit the parasite, whose life cycle depends on successful host development and dispersal [29]. Additionally, prolonged oviposition could expand the spatial reach of the parasite, increasing opportunities for horizontal transmission to unrelated individuals.

While our data suggest the existence of mutualistic traits, we cannot exclude the possibility that some of the observed effects result from host compensatory plasticity. In many species, females increase investment in reproduction when facing threats like parasitism or predation, potentially as an adaptive response to reduce offspring mortality [59–62]. Although we did not observe higher fecundity in parasitized females under normal conditions, differences in oviposition behavior and progeny size could still represent a plastic adjustment to infection. Notably, fecundity impairment occurred only in unparasitized females fed with diluted blood— conditions mimicking reduced resource acquisition associated with host defensive behaviour, which leads to partial feeding in nature [63].

This study was conducted using a single laboratory line of *Ae. albopictus*, either infected or uninfected with its native parasite assemblage. Genetic diversity in this line was likely reduced through long-term laboratory maintenance, although this was not directly measured. Both mosquito and parasite field populations appear to be locally structured [21,64], while also exhibiting genetic structuring at the global scale [21,65,64]. Genotype × genotype interactions have been shown to influence parasite infection and proliferation [24]. Controlling mosquito and parasite genetic diversity could therefore help disentangle the flexibility of these interaction outcomes and should be addressed in future studies. In addition, the nature of this interaction should be investigated for other fitness-related traits and under more complex conditions (e.g., varying ecological contexts) to better assess its consequences for mosquito fitness. Indeed, gregarine–mosquito relationships have been shown to vary with respect to behaviour, development, and survival depending on the context [23,26,66–68]. Finally, although the prevalence of *Ascogregarina taiwanensis* is high at the global scale, it varies among populations and over time [21,26,69].

In summary, our study highlights the complexity of the interaction between *Ae. albopictus* and *As. taiwanensis*, revealing how host physiological responses to infection can offset parasite-associated costs and generate outcomes consistent with mutualistic traits under specific conditions. These traits should be further investigated in relation to other mosquito life-history traits and under more ecologically realistic conditions. The parasite is transmitted through a combination of environmental exposure and maternal bias, and infection is associated with improved blood digestion and nitrogen assimilation during oogenesis. Future studies should focus on tracking nutrient fluxes between host and parasite, particularly toward the ovaries, to further unravel the metabolic interdependence shaping this unusual interaction.

## Materials and methods

### Mosquito lines

Parasitized and unparasitized *Aedes albopictus* lines originated from a hybrid laboratory population derived from two French natural populations sampled in Villeurbanne (N: 45°46′18.990″, E: 4°53′24.615″) and Pierre-Bénite (N: 45°42′11.534″, E: 4°49′28.743″) in 2017, following a previously described protocol [21,70]. Briefly, this population was maintained in large cages (~5,000 individuals). To establish the parasitized line, we applied conditions favoring parasite transmission: (i) water containers were kept inside the cage during oviposition, (ii) larvae were reared in this same water, and (iii) during the first generations, ~ 100 parasitized adults were crushed and added to the water to reseed the next generation and ensure efficient parasite transmission. Parasite presence was verified by diagnostic PCR and microscopy as previously described [70]. To generate the unparasitized line, ~ 10,000 eggs originating from the parasitized line were used, and parasite transmission was deliberately disrupted. Specifically: (i) during the first ten generation, egg surfaces were rinsed and treated with an antiparasitic compound effective against gregarines (griseofulvin, 1 mg mL$^{-1}$); (ii) at each generation, eggs were systematically removed from the water container; and (iii) transferred to fresh water. Thus, both lines share the same genetic background but differ in infection status and were each maintained at ~5,000 individuals. *As. taiwanensis* presence was monitored at each generation ($F_{46}$) by examining crushed pools of 25 adults and samples of rearing water under ×400 magnification using an optical microscope (Leica). Additionally, a subset of 20 parasitized and 20 unparasitized non-fed females was examined during the experiment. The infection rate was 80% (16/20) in the parasitized population, with a mean parasite intensity of 13,125 ± 8,139 oocysts per individual, and 0% (0/20) in the unparasitized population. Because infection prevalence in the parasitized line does not reach 100%, the effects attributed to the parasite in our results are likely conservative and may therefore be underestimated. Previous work demonstrated that *As. taiwanensis* can efficiently recolonize unparasitized populations, indicating no apparent loss of adaptation [70]. Both lines were reared under identical conditions in a biosafety level 2 insectary: 28°C, 80% relative humidity, and an 18h:6h light/dark cycle. Larvae were reared in 1.5 L of dechlorinated water and fed daily with a 25:75 yeast:fish food mixture (Biover; Tetra). Adults were kept in cages and provided with 10% sucrose *ad libitum*.

## Prevalence of *As. taiwanensis* across the female reproductive cycle

To track parasite prevalence throughout the female reproductive cycle, oocyst presence was recorded at key stages (S5 Fig): unmated (UNMAT, 7 days old), mated (MATED, 14 days), 1 day after blood feeding (1DABF, 15 days), and 3 days after blood feeding (3DABF, 17 days). These timepoints correspond to pre-reproductive, host-seeking, early oogenesis, and post-oogenesis stages, respectively. Fifteen females per condition from the parasitized line were individually crushed in 1.5 mL tubes with 100 µL sterile water and three 3 mm glass beads using a FastPrep lysis system (MP Biomedicals; 30 s, 20 m/s). Samples were observed at ×400 magnification to detect oocysts. As a control, ten females from the unparasitized population were examined for infection at the same life stages as parasitized individuals to confirm the absence of oocysts.

## Pseudo-vertical transmission assays

To assess pseudo-vertical transmission, blood-fed females were isolated 2 days post-blood meal in 50 mL tubes with 25 mL sterile water and a piece of blotting paper. After 6 days (end of oviposition), eggs were either kept in the same container (enabling waterborne and egg-smearing transmission) or transferred to a new container with sterile water (enabling only egg-smearing transmission). One milligram of food was added to each tube. Eggs were vacuum-incubated at −20 Hg for 6 h at 28°C. Five days post-hatching, third- and fourth-instar larvae were dissected under a stereomicroscope to detect *As. taiwanensis* trophozoites. Ten larvae per female were examined: 10 females for waterborne (n = 100 larvae), 9 for egg-smearing transmission (n = 90 larvae). Females that died were removed from the analysis. To determine whether oocysts associated with eggs were firmly attached or could be easily detached when eggs are flooded, an additional control experiment was performed. Eggs laid by parasitized females were collected from a large cage containing approximately 5,000 individuals. One hundred eggs were directly hatched in 50 mL of unparasitized freshwater. Another set of 100 eggs was rinsed three times with fresh water and then allowed to hatch in a new container containing 50 mL of unparasitized freshwater. The prevalence of trophozoites in larvae was assessed in 28 individuals from the first condition and 30 individuals from the second condition.

## Blood intake assessment

Unfed and blood-fed females were anesthetized on ice immediately after feeding. Abdomens were photographed at ×40 magnification, and width was measured in ImageJ v2.0 [71] for 15 individuals per condition. Abdomen width served as a proxy for blood intake [72]. Additionally, 261 unfed parasitized females and 254 unfed unparasitized females were photographed as x7.5 magnification, and their abdomen surface was measured in ImageJ v2.0.

## Oogenesis dynamics

To enhance vitellogenesis, 14-day-old females were starved for 6 h, then fed for 20 min using an artificial feeder (Hemotek) containing undiluted or diluted (1:2 or 1:5 with PBS) rat blood. The feeder was covered with pork intestine and maintained at 37°C. Females were dissected immediately (0 h), or 24 h, 48 h, or 72 h post-feeding. For each female, 6–10 primary follicles were photographed at ×100 magnification, and follicle and yolk areas were measured in ImageJ v2.0 [71]. Experiments with diluted blood were performed on 5 females per condition (10 ovaries/female). Females that died were removed from the analysis.

## Egg-laying dynamics, hatching rate, and larval size

Two days post-blood meal, 25 parasitized and 25 unparasitized females were individually placed in 50 mL tubes with 25mL of dechlorinated water and blotting paper. Females were transferred daily to new tubes for 6 days, during which egg numbers were recorded; no eggs were laid beyond this time window. Eggs were either immediately hatched or stored for 2 weeks

on a dry blotting paper at 28°C and 80% relative humidity. To enhance hatching, they were vacuum-incubated (−20 Hg, 8 h, 28°C) with 1 mg fish food to stimulate hatching. For 13 parasitized and 14 unparasitized females, 5–8 first-instar larvae were photographed (×40), and size was measured in ImageJ v.2. [71]. Total larval counts were recorded after one week.

### RNA extraction, library preparation and sequencing

Five parasitized and five unparasitized females were collected from each reproductive condition (*i.e.,* UNMAT, MATED, 1DABM, 3DABM). Prior to euthanasia in liquid nitrogen, individuals were gently stimulated for 10 min to mimic natural activity. RNA was extracted following a published method (56). Libraries were prepared using the TruSeq Stranded mRNA kit (Illumina) and sequenced (2 × 150 bp) on a NovaSeq 6000 (Microsynth AG). Reads (17.7–30.8 million/sample) were demultiplexed and adapter-trimmed. Data are available at ENA (ERS23447863).

### Transcriptome analysis of *Ae. albopictus*

Reads were processed on Galaxy (https://usegalaxy.fr/) following Batut et al. [73]. Quality was assessed with FastQC v0.12.1 [74], trimmed with Cutadapt v4.8 [75] with a minimum length of 75 bp and a minimal quality score of 30, and mapped to the *Ae. albopictus* genome (RefSeq GCF_006496715.1) using STAR v2.7.11a [76]. Mapped reads were counted with featureCounts v2.0.3 [77] and normalized/analyzed with DESeq2 v1.42.1 [78]. DEGs (adj. p < 0.05, Benjamini-Hochberg correction) were annotated via VectorBase. GO-terms were simplified using the R package *simplifyEnrichment* v1.12.0. The KEGG database was used for pathway annotation.

### *De novo* transcriptome analysis *As. taiwanensis*

Unmapped reads were extracted with Samtools v1.15.1 (64) and assembled de novo following the recommendation of Bretaudeau *et al.* [79]. Trinity v2.15.1 [80] was used with the Salmon method [81] for quantification. Transcripts were aligned to 17 Apicomplexan genomes from CryptoDB (https://cryptodb.org/cryptodb/app/, downloaded 12 Dec 2023). *Ascogregarina taiwanensis* is the only Apicomplexan species known to be part of the *Ae. albopictus* microbiota [21] and the CryptoDB database includes *Gregarinasina* as well as phylogenetically close taxa. A total of 4,676 transcripts were retained as *As. taiwanensis* and deposited at Zenodo (https://zenodo.org/records/18374963) DEG analysis followed the same pipeline as for mosquitoes, and annotations (GO-terms and KEGG pathways) were performed using CryptoDB.

### Quantification of dry mass, protein, uric acid and uricase activity

Ten pools of three females per condition were freeze-dried (8 h) and weighed. Samples were homogenized in 200 µL cold PBS (pH 7.4) with three 3 mm beads using a FastPrep-24 system (30 s, 6 m/s), centrifuged (1000 g, 5 min), and supernatants were transferred to black 1.5 mL microtubes (Heatrow Scientific). Protein content was measured using the Qubit Protein Assay Kit and a Qubit4 fluorometer (Invitrogen). Uric acid and uricase activity were quantified using the Amplex Red Kit (Invitrogen), with fluorescence measured at 570 nm (excitation at 530 nm) on a SpectraMAX iD3 plate reader (Molecular Devices). Data were converted using standard curves.

### Statistical analysis

All analyses were conducted in R v4.4.0 [82]. Female ID was used as a random effect in mixed models. Parasite prevalence over time and larval prevalence were analyzed via binomial GLMMs using the glmmTMB package or a Fisher exact test. Abdomen width data were transformed (Box-Cox) and analyzed using linear models (*stats*, *MASS*). Egg chamber, yolk area, and larval size were assessed with linear mixed models (*lme4*). Egg-laying dynamics were modeled with Poisson GLMMs (*glmmTMB*). Hatching rate was analyzed via binomial GLMMs (*lme4*). Protein, uric acid, and uricase data were normalized to dry mass and analyzed using linear mixed-effects models (*lme4*). Model assumptions were assessed using graphical inspection and simulation-based residual diagnostics (*DHARMa*). Significance was tested using type II

ANOVA or Wald Chi-square tests (*car*), followed by Tukey-HSD *post hoc* tests (*emmeans*). PCA and RDA (with permutation tests) of transcriptomic data were performed using *DESeq2* and *vegan.* Datasets and R scripts are available at Zenodo (https://zenodo.org/records/18374963).

## Supporting information

**S1 Fig. Egg smearing validation essay.** The proportion of infected L4 larvae was estimated after hatching from eggs transferred directly into fresh water (Non washed eggs) or from washed eggs transferred into fresh water (Washed eggs). The number of screened larvae is indicated for each condition (n). Statistical significance was assessed using Fisher's exact test ($p = 0.048$).
(PDF)

**S2 Fig. Abdomen area of parasitized and unparasitized females.** The area was measured on unfed females from parasitized and unparasitized mosquito lines. The value is expressed in mm$^2$. Welch's t-test p-value = 0.0253.
(PDF)

**S3 Fig. Dynamics of egg laying.** The number of eggs laid by each female was recorded daily and reported for each day of the oviposition period.
(PDF)

**S4 Fig. Impact of *As. taiwanensis* and blood quality in oogenesis and egg laying.** (a) The follicle primary chamber and (b) yolk areas are reported for parasitized and unparasitized females 1DABM female mosquitoes using 1/2 or 1/5 diluted blood. (c) The number of eggs laid in each of those conditions was reported. Asterisks represent significant pairwise differences at a threshold of $p \leq 0.05$ from (b) post hoc Tukey HSD and (c) Wilcoxon rank-sum tests.
(PDF)

**S5 Fig. Overview of the experimental design.** *Ascogregarina taiwanensis* prevalence was assessed in *Ae. albopictus* females at four life stages: unmated (UNMAT), mated (MATED), one day after blood meal (1DABM), and three days after blood meal (3DABM). Parasite transmission to offspring was evaluated via water and egg smearing. Blood intake was quantified by measuring abdomen width before and after feeding. Oogenesis was monitored by tracking primary follicle and yolk development over time. Egg-laying dynamics and larval size were recorded. Comparative transcriptomic analyses were performed to assess both parasite gene expression and its impact on the mosquito transcriptome. Protein content, uric acid levels, and uricase activity were also measured at each stage. Biorender was used to generate the figure (Licence number TW29FC9MJM). Created in BioRender. Girard, M. (2026) https://BioRender.com/lyuz2tc.
(PDF)

**S1 Table. Metabolic pathways identified with the DEGs of 1 one day after blood meal (1DABM) parasitized females.**
(XLSX)

**S2 Table. Metabolic pathways identified in the As. taiwanensis transcriptome at different stages of the female life cycle, i.e., unmated (UNMAT), mated (MATED), one day after blood meal (1DABM) and three days after blood meal (3DABM).**
(XLSX)

## Acknowledgments

We also acknowledge the IBIO bioinformatics platform for assistance with RNA-seq analyses. As none of the authors are native English speakers, the final draft of this manuscript was reviewed with the help of artificial intelligence software based on large language models to improve clarity. All suggested modifications were carefully checked by the authors to ensure that the meaning was not altered.

## Author contributions

**Conceptualization:** Maxime Girard, Guillaume Minard.

**Data curation:** Maxime Girard, Mathieu Laÿs, An-nah Chanfi, Guillaume Minard.

**Formal analysis:** Maxime Girard, Mathieu Laÿs, Guillaume Minard.

**Funding acquisition:** Guillaume Minard.

**Investigation:** Maxime Girard, Mathieu Laÿs, Edwige Martin, Laurent Vallon, An-nah Chanfi, Melanie Bretton, Aurélien Vigneron, Séverine Balmand, Guillaume Minard.

**Methodology:** Maxime Girard, Mathieu Laÿs, Edwige Martin, Laurent Vallon, An-nah Chanfi, Melanie Bretton, Aurélien Vigneron, Séverine Balmand, Guillaume Minard.

**Project administration:** Anne-Emmanuelle Hay, Claire Valiente Moro, Guillaume Minard.

**Resources:** Guillaume Minard.

**Supervision:** Patricia Luis, Anne-Emmanuelle Hay, Claire Valiente Moro, Guillaume Minard.

**Validation:** Maxime Girard, Claire Valiente Moro, Guillaume Minard.

**Visualization:** Maxime Girard.

**Writing – original draft:** Maxime Girard, Guillaume Minard.

**Writing – review & editing:** Maxime Girard, Aurélien Vigneron, Patricia Luis, Anne-Emmanuelle Hay, Claire Valiente Moro, Guillaume Minard.

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
