## [Decision Letter · Decision Letter 0]

8 Dec 2025

Mutualism in disguise: a mosquito parasite with mixed transmission mode displays mutualistic traits promoting oogenesis.

PLOS Pathogens

Dear Dr. Minard,

Thank you for submitting your manuscript to PLOS Pathogens. After careful consideration, we feel that it has merit but does not fully meet PLOS Pathogens's publication criteria as it currently stands. Therefore, we invite you to submit a revised version of the manuscript that addresses the points raised during the review process.

We look forward to receiving your revised manuscript.

Kind regards,

Dominique Soldati-Favre

Section Editor

Editor-in-Chief

PLOS Pathogens

orcid.org/0000-0003-2946-9497

Editor-in-Chief

PLOS Pathogens

orcid.org/0000-0002-7699-2064

**Additional Editor Comments:**

This manuscript is strong and addresses a highly relevant question in evolutionary biology, shedding light on the continuum from parasitism to mutualism. The combination of experimental and sequencing approaches provides a valuable contribution to understanding this host–parasite system.

However, the revision must clarify the genetic homogeneity of both host and parasite lines, as this is essential for interpreting transmission patterns. Additionally, the authors should verify the parasitization status of females and provide evidence if they wish to maintain the claim regarding egg-smearing as a vertical transmission route. Finally, all specific comments raised by the reviewer to improve clarity should be addressed.

**Journal Requirements:**

1) Please provide an Author Summary. This should appear in your manuscript between the Abstract (if applicable) and the Introduction, and should be 150-200 words long. The aim should be to make your findings accessible to a wide audience that includes both scientists and non-scientists. Sample summaries can be found on our website under Submission Guidelines:

https://journals.plos.org/plospathogens/s/submission-guidelines#loc-parts-of-a-submission

Potential Copyright Issues:

- Figures 3, 4, 6, and S2. Please confirm whether you drew the images / clip-art within the figure panels by hand. If you did not draw the images, please provide (a) a link to the source of the images or icons and their license / terms of use; or (b) written permission from the copyright holder to publish the images or icons under our CC BY 4.0 license. Alternatively, you may replace the images with open source alternatives. See these open source resources you may use to replace images / clip-art:

**Reviewers' Comments:**

Reviewer's Responses to Questions

**Part I - Summary**

Reviewer #1: This very interesting manuscript addresses the evolutionary dynamics of mutualism in symbionts that exhibit both vertical and horizontal transmission, focusing on the mosquito parasite Ascogregarina taiwanensis. To my understanding, vertical transmission, pseudo-vertical transmission, and horizontal transmission, when occurring between genetically related individuals in a homogeneous host population with a homogeneous parasite population, may lead to the same outcome: the evolution of mutualistic traits in both host and parasite. The reason lies in the successful adaptation of the host to the parasite, as horizontal transmission in this context occurs between hosts already adapted to the particular genotype of the parasite. If horizontal transmission were to occur between distantly related hosts, such as different mosquito populations, or if distantly related parasites were to infect the host, the negative effects would very likely outweigh the positive ones.

From my perspective, every symbiotic relationship is essentially parasitic, but in some cases, the host can benefit, usually after mutual adaptation. This is because a symbiotic relationship typically begins as parasitic or pathogenic, representing the most unbalanced form of symbiotic interaction. The mutualistic state results from adaptation or more effective defence mechanisms in the host, which may eventually derive some advantage from the originally parasitic relationship. Of course, vertical transmission allows for the maintenance and strengthening of adaptation, thus fostering a mutual relationship. However, in my opinion, the same effect can be achieved by horizontal transmission within genetically homogeneous populations of parasites and hosts, and this manuscript supports that view. In other words, the mode of transmission is less important than the level of adaptation (in experiments, the genetic homogeneity) of both parasites and hosts. Such genetic traits can be found exclusively in vertically transmitted symbionts, but horizontal transmission itself does not necessarily make the symbionts less adapted to the host and/or more genetically diverse. This is underlined by the strict host specificity of gregarines, as these symbionts/parasites, being early-branching and evolutionarily ancient Apicomplexa, may prefer well-adapted hosts (and accumulating evidence suggests this) as a more sustainable reproductive and evolutionary strategy.

In many cases, such as A. taiwanensis infections, the mutualistic positive effect is actually a kind of side effect; the parasite causes elevated transcription of genes involved in development, nitrogen, and protein metabolism, likely to compensate for the uptake of nutrients by the parasite, leading to an extended laying period and larger eggs and larvae, as a “collateral” advantage for the host. As this compensates for the negative effects of A. taiwanensis presence in the mosquito host, such as the fitness decrease of A. albopictus through reduced female fecundity, slower larval development, and increased larval mortality, which have been referred to elsewhere, the presence of the parasite does not place the host at an evolutionary disadvantage, thereby stabilising the symbiotic relationship.

Reviewer #2: Girard et al. is an extremely interesting study of parasites in the genus Ascogregarina, which infect certain species of mosquitoes (most notably in the genus Aedes). The study finds evidence of mutalistic traits, although it does not measure every component that would be required to strictly define the interaction as a mutualism (the authors do not claim to do so, either). The experimental design is interesting, but there are key details of methodology missing, most critically in how mosquitoes were exposed to parasites, and how mosquitoes were confirmed as infected. The manuscript itself is extremely thorough, and the results are noteworthy and of general interest to parasitologists, assuming the authors can clarify methods. The combination of experimental study with sequencing brings considerable strength to the work.

Although I feel the conclusions are overall strong, I have a few general comments below:

-I have general concerns about the status of mosquitoes as being defintively parasitized when exposed to parasites. The authors own figures (e.g. Fig 1 B) shows that ~20-25% of exposed females are not infected despite being exposed, unless taking a bloodmeal clears parasites (does not seem so). Did the authors confirm that mosquitoes they label as "parasitized" were indeed parasitized, or is it that these are mosquitoes from the lab colony in which parasites are also amplified? At present, it appears that the label of parasitized is applied for mosquitoes exposed to parasites, but we do not know if these mosquitoes are all parasitized. If they are not all parasitized, some of the effects the authors attribute to parasitism may be instead a result of some individuals experiencing a competitive advantage over infected conspecifics, as the authors point out that infection reduces competitive ability in larval stages (which has direct outcomes on adults). The subtantially higher IQDs seen in some Fig2 boxplots could be a result of variable parasite load and/or some females being uninfected entirely.

-Similarly, the infection dynamics of As. taiwanensis in Ae. albopictus have been strongly associated with different parasite exposure levels. It is unclear from the methods what the exposure level that was used here was. This is important information to be present in the current manuscript, even if some details are in previous publications, as it allows a more clear comparison between these and past results.

-I am not entirely sure I agree that egg smearing is vertical transmission. Egg smearing does not mean that the offspring will be infected, as eggs of most Aedes with Ascogregarina must be submerged prior to hatching, which will release oocysts into the environment for consumption by any larva present, not only the offspring of the ovispositing female. I comment on this more below.

-The authors discuss mutualism at length, but do not assess sufficient components of fitness to clearly conclude that this is a mutualism. While they present compelling evidence on how As. taiwanensis may have some mutualistic characteristics, it is unclear if these results are overall strong enough to counteract previously described effects of the parasite. See comments below.

Despite these concerns, this manuscript contains extremely interesting results and is overall well-written. I want to emphasize this, as though my above comments are critical, I think there is a lot of excellent work here.

Section by section feedback, below:

Introduction

Overall quite thorough.

L71: There is some evidence that other characteristics than just nutrient amounts may impact host fitness - although in another species of Ascogregarina, Sulaiman (1992) demonstrated that some strains are more pathogenic to some populations of host, which may indicate a degree of adaptation of different populations. Tseng (2017) provides further evidence of this with As. taiwanensis and Ae. albopictus due to the apparent "escape" from parasitism by hybrid Ae. albopictus. In the field, various authors report quite varying infection intensity (sometimes seasonal), and Soghigian and Livdahl (2021) shows some degree of population regulation. While field larvae may be under nutrient defecient conditions broadly, perhaps those are the more natural condition of hosts.

L81: Interesting distinction between horizontal and vertical here. Just a comment, rather than something to address, but anecdotally it seems container-dwelling mosquitoes will revisit the same container they emerge from to lay eggs; in this case, horizontal and vertical transmission modes end up overlapping. I am also not sure that I would agree that

L81: Hm, the more I think about this, the less certain I am that this is indeed vertical transmission. Mosquitoes often develop in larval habitats with unrelated conspecifics, other mosquito species, and other aquatic organisms. Many of these organisms may consume the parasite material, such that it is not, strictly speaking, vertical transmission. I would instead think that egg smearing increases the probability of transmission from mother to offspring, but in the same way that defecation and/or dying on the water and/or reusing a larval habitat a female emerged from would. Do we know that e.g. egg smearing increases the chance that larvae will be infected, or increases the parasite load?

Results

I just want to comment that, despite various feedback, the experimental design was really interesting and the results are clearly articulated and easy to follow.

L110: Did the authors observe female oviposition behavior at all? Were all females left to die in the water after oviposition? There are some interesting implications here, as at least in Ae. aegypti, females contact water during oviposition just with their feet ("tasting" it) rather than directly on it, so it is possible that egg smearing may be a primary way that a skip ovipositing species distributes Ascogregarina, unless the female was also defecating during oviposition.

L125: Really interesting observation on the shift in egg-laying. Do the authors have any estimate on adult size for ovipositing females? Were parasitized females smaller? I wonder if this could explain this shift - may be worth mentioning, as there is evidence that Ascogregarina reduce adult size I believe.

L128: Hm, Figure 2E certainly looks like there were fewer eggs laid by parasitized females, though I recognize the difference is not significant. The authors may want to present the average per female per day here, from the 25 females, rather than the total count.

L130: Given parasitized mosquitoes are generally smaller, and given that females can 'pass on' their infection, I wonder if this minor size advantage at first instar will actually translate through the life cycle.

As I am not an expert in transcriptomics or GO enrichment analyses, I will limit my commments from these sections. As written, they are easy to understand and do present strong evidence of the effects of parasitism on Ae. albopictus.

L218: This is a very interesting experimental design... But what size were females that were used here? It seems possible that if parasitized females were smaller, they may be able to make use of less bloodmeal resources than larger females, given that the bloodmeal is also used for host nutrition (albeit in a minor way). In addition, when looking at supplemental figures... It appears that unparasitized females still had egg development, just not oviposition.

Discussion

L257: I don't agree with many of these statements, based on my understanding of the author's experimental design. From what I understand, mosquitoes were transferring to oviposition environments basically starting on Day 3. Then, both sets of females oviposited primarily over a two day period; it is just that infected females took an extra day for to start ovipositing in earnest. As females were not given access to oviposition sites before this, we don't know if, e.g., unparasitized females may have begun earlier than parasitized if given the option. Since the primary pattern appears to be a shift to a day later in parasitized females, this indicates a longer time until oviposition, NOT an extended period of oviposition / more skip oviposition. As such, I do not see the pattern the authors report of an increased length of oviposition - it looks like all treatements continued to lay a small number of eggs after day 4, as both boxplots shift slightly upwards.

L272: I would limit this mutualism commentary to specifically what the authors have measured. We have no idea of how large females are, or how large parasitized larvae are in habitats with conspecifics, only at first instar. We do not know the survival of such larvae, etc. The authors have certainly demonstrated mutalistic traits; however, they have not definitively shown a mutualism as we do not know the ultimate fitness consequences of parasite infection from this study. Interestingly, there is another study with implicated mutualist traits in Ascogregarina (Soghigian et al. 2017), though it appears it is in a different species of parasite and Aedes host.

L288: This is extremely interesting, but it is unclear how oocysts would acquire glutamine while in that stage.

L302: While skip oviposition most certainly is a valuable bet hedging strategy, this statement reads as if to suggest As. taiwanensis is triggering skip oviposition - it does not seem so. Rather, there seems to be delayed oviposition,

L316: I was under the impression the study the authors cite has to do with behavioral changes in response to partial bloodmeals, not diluted bloodmeals. Moreover, adult female size seems confounding here, as it is quite possible that parasitized females were smaller than unparasitized females and able to better utilize partial bloodmeals.

Methods

Did the authors ensure females classified as "parasitized" were all, in fact, parasitized?

L337: Was only presence recorded, or was infection intensity also recorded? There appears to be dosage-related effects of As. taiwanensis from the literature (Sulaiman 1992 and others).

L349: What happened to females after they finished ovipositing? Did they die in the water? Minor - it does not impact the robustness of the vertical transmission assay - but worth noting if so.

L366: How long were females with diluted blood given to oviposit?

L430: Did the authors evaluate assumptions for their ANOVA? Some of the boxplots suggest distributions that may not be homoscedastic, but it is hard to tell from figures.

**Part II – Major Issues: Key Experiments Required for Acceptance**

Reviewer #1: I have only one major concern, which lies in the missing description of the genetic homogeneity of the experimentally used hosts and parasites. While it is mentioned in the Materials and Methods section that lines of infected and uninfected A. albopictus were used, it is not clear to me whether these two lines are genetically identical (and likely have the same level of adaptation to the presence of the parasite) and the presence or absence of the parasite is the only difference between them. Similarly, I have not found any information concerning the genetic structure of the parasite, which I am convinced is essential for a proper understanding of vertical and horizontal transmission, as genetic homogeneity is, in my opinion, more important than the mechanical mode of transmission.

Reviewer #2: -My major concern is with regards to the status of parasitized females; an experiment may be required to demonstrate that females that are labeled as parasitized are all indeed parasitized.

-Additionally, the authors refer to the idea of egg-smearing as vertical transmission. I'm not certain this claim is central to their manuscript, but one way to show this would be to demonstrate that larvae exposed to egg-smearing and other conditions have a higher parasite prevalence and/or parasite load than those NOT egg-smeared (e.g. rinsed eggs or eggs from a non-parasitized colony.

**Part III – Minor Issues: Editorial and Data Presentation Modifications**

Reviewer #1: (No Response)

Reviewer #2: -The authors should provide additional caveats to their conclusions regarding a mutualism. These interactions are measured only in the lab, under relatively low resource stress conditions, and measure only a select few components.

-I am unconvinced that egg-smearing is truly vertical transmission, and I'm not sure that is required for interpretation of the results the authors present. Demonstration of egg-smearing is an important observation, as it has been assumed to occur in the literature without apparent justification.

PLOS authors have the option to publish the peer review history of their article (what does this mean? ). If published, this will include your full peer review and any attached files.

**Do you want your identity to be public for this peer review?** For information about this choice, including consent withdrawal, please see our Privacy Policy .

Reviewer #1: No

Reviewer #2: No

**Figure resubmission:**

**Reproducibility:**



---

## [Decision Letter · Decision Letter 1]

24 Feb 2026

Dear Dr. Minard,

We are pleased to inform you that your manuscript 'Mutualism in disguise: a mosquito parasite with mixed transmission mode displays mutualistic traits promoting oogenesis.' has been provisionally accepted for publication in PLOS Pathogens.

Best regards,

Dominique Soldati-Favre

Section Editor

PLOS Pathogens

Sumita Bhaduri-McIntosh

Editor-in-Chief

PLOS Pathogens

orcid.org/0000-0003-2946-9497

Michael Malim

Editor-in-Chief

PLOS Pathogens

orcid.org/0000-0002-7699-2064

Reviewer Comments (if any, and for reference):

Reviewer's Responses to Questions

**Part I - Summary**

Reviewer #1: The authors addressed my concerns and comments to my satisfaction. I recommend that this manuscript be accepted by PLoS Pathogens.

Reviewer #2: The authors have, overall, addressed my feedback quite thoroughly. I am no longer concerned with their wording or descriptions they use in the manuscript, e.g. associated with vertical vs horizontal transmission or with mutualism vs mutalistic traits. I see this work as a valuable contribution to our understanding of host-parasite interactions and it presents initial evidence of a complex interaction in a system classically defined as either a commensalism or a weak parasitism, although in some closely related organisms/parasites, there has already been some implication of mutalistic interactions.

I will note, that the authors say that they believe that inclusion of unparasitized females in parasitized groups would underestimate the effect of parasitism, but this may not actually be the case - for instance, if 25% of the parasitized group effectively experienced an intraspecific competitive advantage due to escape from parasitism, it is possible some components attributes to mutualistic interactions may not be so, particularly related to maternal partioning. We know the competitive ability of infected Ae. albopictus is lower than uninfected Ae. albopictus, at least in interspecific interactions. However, evaluating whether or not unparasitized females are responsible for any amount of the findings of the authors would require significant additional work that I do not think is necessary at this stage.

Despite this, I do think the authors have done a commendable job addressing my previous concerns, including some new experiments and/or clarifying details that greatly assisted in my understanding of their analyses. I believe the manuscript is now substantially clearer and stronger, and details an interesting interaction that highlights the complexity of parasite and host interactions.

**Part II – Major Issues: Key Experiments Required for Acceptance**

Reviewer #1: The authors addressed my concerns and comments to my satisfaction. I recommend that this manuscript be accepted by PLoS Pathogens.

Reviewer #2: Teh authors addressed my key concerns.

**Part III – Minor Issues: Editorial and Data Presentation Modifications**

Reviewer #1: The authors addressed my concerns and comments to my satisfaction. I recommend that this manuscript be accepted by PLoS Pathogens.

Reviewer #2: The manuscript addressed my prior minor issues.

PLOS authors have the option to publish the peer review history of their article (what does this mean? ). If published, this will include your full peer review and any attached files.

**Do you want your identity to be public for this peer review?** For information about this choice, including consent withdrawal, please see our Privacy Policy .

Reviewer #1: No

Reviewer #2: No

---

## [Editor Report · Acceptance letter]

Dear Dr. Minard,

We are delighted to inform you that your manuscript, "Mutualism in disguise: a mosquito parasite with mixed transmission mode displays mutualistic traits promoting oogenesis.," has been formally accepted for publication in PLOS Pathogens.

Best regards,

Sumita Bhaduri-McIntosh

Editor-in-Chief

PLOS Pathogens

orcid.org/0000-0003-2946-9497

Michael Malim

Editor-in-Chief

PLOS Pathogens

orcid.org/0000-0002-7699-2064